# Neoadjuvant Chemotherapy-Chemoradiation for Borderline-Resectable Pancreatic Adenocarcinoma: A UK Tertiary Surgical Oncology Centre Series

**DOI:** 10.3390/cancers14194678

**Published:** 2022-09-26

**Authors:** Rachna Gorbudhun, Pranav H. Patel, Eve Hopping, Joseph Doyle, Georgios Geropoulos, Vasileios K. Mavroeidis, Sacheen Kumar, Ricky H. Bhogal

**Affiliations:** 1Department of Surgery, The Royal Marsden Hospital, Fulham Road, London SW3 6JJ, UK; 2Upper Gastrointestinal Research Group, Department of Radiotherapy, Institute of Cancer Research, 123 Old Brompton Road, London SW7 3RP, UK

**Keywords:** pancreatic cancer, neoadjuvant treatment, Whipple’s surgery, pancreatic resection

## Abstract

**Simple Summary:**

Treatment of pancreatic cancer with chemotherapy followed by chemoradiotherapy prior to surgery in patients where the tumour is in contact with major abdominal blood vessels improves the ability to completely resect the tumour. This, in turn, improves patient survival after surgery, demonstrating that this treatment strategy is appropriate for such tumours.

**Abstract:**

Background: Patients with borderline-resectable pancreatic ductal adenocarcinoma (BR-PDAC) have historically poor survival, even after curative pancreatic resection and adjuvant chemotherapy. Emerging evidence suggests that neoadjuvant chemoradiation (NCR) improves R0 resection rates in BR-PDAC patients. We evaluated the R0 resection rate, disease-free survival (DFS) and overall survival (OS) in our patients who underwent NCR for BR-PDAC at our institution. Methods: All patients who underwent NCR for BR-PDAC from January 2010 to March 2020 were included in the study. The patients received a variety of NCR regimens during the study period, and in patients with radiological evidence of tumour stability or regression, pancreatic resection was performed. The primary endpoint was the OS, and the secondary endpoints included patient morbidity, the R0 resection rate, histological parameters and the DFS. Results: The study included 29 patients (16 men and 13 women), with a median age of 65 years (range 46–74 years). Of these 29 patients, 17 received FOLFIRINOX and 12 received gemcitabine (GEM)-based NCR regimens. All patients received chemoradiation at the end of chemotherapy (range 45–56 Gy). R0 resection was achieved in 75% of the patients, with a higher rate noted in the FOLFIRINOX group. The median DFS was 22 months for the whole cohort but higher in the FOLFIRINOX group (34 months). The median OS for the cohort was 29 months, with a higher median OS noted for the FOLFIRINOX cohort versus the GEM cohort (42 versus 28 months). Conclusion: NCR, particularly FOLFIRINOX-based treatment, for BR-PDAC results in higher rates of R0 resection and an increased median DFS and OS, supporting its continued use in this patient group.

## 1. Introduction

Pancreatic ductal adenocarcinoma (PDAC) remains associated with disappointing, poor long-term patient survival, with reported rates of 5–10% patient survivorship 5 years following surgery [1,2,3]. Traditionally, PDAC has been divided into two distinct surgical categories, resectable disease, where the tumour demonstrates no radiological vascular involvement, and locally advanced disease, where the tumour shows arterial and/or major venous involvement. However, more recently, the term ‘borderline-resectable (BR)’ has been used to describe PDAC that is potentially resectable but is associated with a degree of vascular involvement/contact, rendering an increased risk of a positive margin (R1) after surgical resection [4]. Surgery with clear resection margins (R0) offers the only curative option for PDAC. In patients with resectable disease, surgery with adjuvant treatment is the recommended pathway [2]. In patients with pre-operative venous and/or arterial involvement, the pathway can be targeted to specific patient and/or tumour characteristics [5]. In 2019, the National Comprehensive Cancer Network (NCCN) updated its classification for PDAC [6]. Here, BR-PDAC is classified by reconstructable venous involvement of the superior mesenteric vein (SMV) or the portal vein (PV), along with abutment but not encasement of arterial structures or removable retroperitoneal structures [7]. Upfront surgery in patients with BR-PDAC is associated with higher rates of R1 resection [6]. Based on these observations, authors have advocated that patients with BR-PDAC be offered neoadjuvant treatment to increase the rates of R0 resection, although initially, this strategy was based on retrospective data.

It has been advocated that PDAC be viewed as a systemic disease in the same manner as oesophago-gastric cancer [8] as in the majority of patients with resectable PDAC, PDAC recurs within 2 years of upfront surgery [9]. Indeed, there is a growing feeling in the field that pancreatic cancer be treated in a similar paradigm to oesophago-gastric cancer. Neoadjuvant chemoradiotherapy (NCR) would be expected to reduce the risk of positive surgical margins, the rate of local recurrence and metastatic progression in PDAC patients [7]. The other advantages of NCR include treating a larger pool of PDAC patients, treating potential micro-metastatic disease, restaging prior to surgery and avoiding unnecessary surgery in patients with aggressive tumour biology [10]. Accordingly, in a series of 160 patients with BR-PDAC who underwent NCR, Katz et al. reported a 39% R0 resection rate [11], similar to that reported by Arvnold et al. [12]. The aim of this study was to evaluate the disease-free survival (DFS) and overall survival (OS) for patients undergoing surgery for BR-PDAC treated with NCR at our institution.

## 2. Methods

This retrospective case series included all patients undergoing pancreaticoduodenectomy following chemotherapy and subsequent chemoradiotherapy from January 2010 to March 2020 for BR-PDAC performed at our institution. The treatment strategy for every patient was discussed and validated based on clinical data and cross-sectional imaging by a dedicated pancreatic multidisciplinary team (MDT). The primary endpoint of the study was overall survival (OS), whilst the secondary endpoints were disease-free survival (DFS), patient complications and R0 resection rates. The study was approved by the Institutional Review Board of the Royal Marsden NHS Foundation Trust.

### 2.1. Initial Staging

All pancreatic masses were either cytologically or histologically proven to be consistent with PDAC by endoscopic ultrasound-guided fine-needle aspiration or biopsy or endoscopic retrograde cholangiopancreatography. It was mandatory to have confirmation of PDAC prior to patients commencing any neoadjuvant treatment. All patients had pre-operative biliary drainage achieved by the placement of a metal biliary stent. In our institution, metal biliary stents were preferred to ensure that the risk of biliary stent occlusion or blockage was minimised, such that the patient could complete the neoadjuvant regimen prior to potential surgical resection. In our patient cohort, there were no clinically significant events noted as a result of pre-operative biliary stenting.

Tumours were classified by the pancreatic MDT team based on CT scan images with arterial and portal contrast infusion and met the criteria for BR-PDAC based on 2019 NCCN guidelines. All patient scans were reviewed by an HPB radiologist (GB) to ensure and verify that all cross-sectional images and tumours fulfilled the aforementioned NCCN criteria. In addition, most patients had undergone CT–positron emission tomography (PET) imaging at initial staging to exclude metastatic disease. Post-chemoradiotherapy staging was assessed by a CT scan. Liver MRI was performed only in the case of suspicion or indeterminate liver lesions to exclude metastasis.

### 2.2. NCR Regimen

Due to the length of period covered by the study, the chemotherapy regimen (see Table 1) and the radiotherapy delivery technique varied over time. The chemotherapy regimen used for patients was individualised by the MDT, but in general, the early phase of the study was characterised by the use of gemcitabine-based treatments and the later phase (2017 onwards) was characterised by the use of FOLFIRINOX-based treatments. Our treatment strategy consisted of first-line chemotherapy and, in the absence of progression, chemoradiation. The radiation dose was 54 Gy in 30 fractions with concurrent capecitabine (1600 mg/m^2^ on days of radiotherapy). From 2010–2014, radiation was CT-planned 3D conformal radiotherapy (3DCRT) delivered as either a single phase or a two-phase technique in select cases (phase 1 45 Gy in 30 fractions followed by 9 Gy in a 5-fraction tumour boost). From 2012, 4D CT was implemented to individualise margins required for tumour motion. From 2014, 3DCRT was replaced by intensity-modulated radiotherapy (IMRT) delivery using volumetric modulated arc therapy (VMAT) at a dose of 54 Gy in 30 fractions. Reassessment was carried out by the MDT at the end of first-line chemotherapy and 4–6 weeks after the end of chemoradiotherapy. This reassessment was based on clinical signs and performance status, CA 19–9 levels and CT scan evaluation in order to decide whether to proceed with surgical resection. Surgery was indicated in the case of tumour response or stability, a performance status of 0 or 1 and a decrease in vascular abutment or even in the case of persistent vascular abutment images if limited to less than 180^o^ encasement as per NCCN guidelines. Adjuvant chemotherapy (gemcitabine) was proposed on the MDT decision alone in the case of poor pathological features (R1 resection or major nodal involvement) and carried out depending on the patient’s performance status.

### 2.3. Surgical Procedures and Pathological Protocol

Following the completion of NCR, the patients were restaged with a CT scan and in some cases with CT-PET and liver MRI. In the presence of tumour stability or tumour regression and the absence of metastatic disease, the patients were considered for pancreatic resection. For all patients, pancreatic resection was planned 6 weeks following the end of chemoradiation. During this 6-week period, the patients underwent cardiopulmonary exercise testing (CPEX) to ensure that they were suitable for major pancreatic resection, with an anaerobic threshold of 7 mL/min/kg being deemed satisfactory.

For all BR-PDAC patients, pancreatic resection was carried out using an open surgical approach. In general, a midline incision was used and metastatic disease excluded on exploratory laparotomy. Following this, a Cattell-Braasch manoeuvre was performed. Our preference was to use the ‘artery-first approach’, with either a posterior or a medical approach [13]. Following this, the superior mesenteric artery (SMA) was dissected free of the tumour using dissection in the divestment plane [14]. Once the SMA was freed from the tumour and/or retropancreatic tissue, the portal vein and superior mesenteric vein (SMV) were isolated to allow for proximal and distal venous control and to aid vascular resection, if required [15]. Pancreatic resections were carried out either in the form of classical Whipple’s surgery or pylorus preserving pancreaticoduodenectomy (PPPD), depending on intra-operative findings. Lymphadenectomy was performed as recommended by the International Study Group of Pancreatic Surgery (ISGPS) [16]. When vascular abutment or involvement of the SMV and/or PV was found, a wedge or segmental resection was included in the procedure to achieve oncological clearance. For segmental resection shorter than 4 cm, end-to-end venous anastomosis without a graft was performed. In our reported series, segmental resections longer than 4 cm were not required. From 2018 onwards (*n* = 19), patients had intra-operative frozen sectioning of the pancreatic resection margin performed to exclude tumours at the resection margin. Pancreatic anastomosis was performed either by using the duct-to-mucosa technique or by dunking pancreaticojejunostomy, with intra-abdominal drainage achieved with two 30-Fr Robinson drains. Drain fluid amylase (DFA) was measured on postoperative day 1 in most cases (*n* = 18). Surgical morbidity was defined as significant surgical postoperative complications of Grade III, IV or V, as classified by Dindo et al. [17].

A macroscopic pathological examination of the resected specimen followed using a standardised protocol by serial slicing of the pancreatic head in a single axial plane, perpendicular to the longitudinal axis of the duodenum, to obtain slices covering the tumour and its ranges up to the inked margins. A histopathological response was abstracted from synoptic pathology reports according to the Royal College of Pathology, United Kingdom [18]. R0 resection was defined as a margin strictly superior to 0 mm. R1 resection were defined as tumour cells on the inked margin. Complete response (CR) was defined as no viable cancer cells identified in the primary tumour or lymph nodes. Nodal status was defined as follows: N0, no nodal regional lymph node involvement; N1, 1–3 regional lymph nodes containing cancer; and N2, 4 or more regional lymph nodes containing cancer.

### 2.4. Follow-Up

Patients had follow-up visits with laboratory evaluation every 3 months and CT scans every 6 months for the first 2 years, visits with laboratory evaluation every 3 months and an annual CT scan for year 3 and visits with laboratory evaluation every 6 months and an annual CT scan for years 4 and 5. Additional evaluations prompted by symptoms, the results of laboratory tests or the treating clinician’s discretion were also used to score events.

### 2.5. Statistical Analysis

Kaplan-Meier curves were used to analyse the OS and DFS. Changes in the OS and DFS between the FOLFIRINOX and GEM chemotherapy groups were examined using a log-rank test. Statistical significance was defined by two-sided *p*-value < 0.05. Log-rank analysis and plots were performed using R studio software (version 1.3.1073).

## 3. Results

### 3.1. Patient Demographics

Between January 2010 and March 2020, 29 patients (16 men and 13 women) with a median age of 65 years (range 47–85) were included in the reported study (Table 2). All patients had BR-PDAC based on the NCCN classification, as assessed by a specialist HPB radiologist. In total, 11 patients had pancreatic head PDAC and 18 patients had uncinate process PDAC. All patients had received neoadjuvant chemotherapy (Table 1) followed by chemoradiation and were deemed suitable for pancreatic resection following a review of post-NCR cross-sectional CT imaging at a specialised pancreatic MDT discussion and had undergone satisfactory anaesthetic assessment.

### 3.2. NCR Regimen

Analysis of our departmental database demonstrated that from January 2015 to March 2020, 62 patients received neoadjuvant chemotherapy followed by chemoradiation for histologically/cytologically proven BR-PDAC. Of these, 17 patients were deemed to have a tumour that was radiologically stable or that had responded to NCR, resulting in a conversion rate to surgery following an NCR rate of 27%. Specifically of these 17 patients, 13 patients demonstrated a response to treatment and 4 patients demonstrated stable radiological tumour characteristics. Of the remaining 45 patients who did not undergo surgery, 11 patients declined surgical intervention, 16 patients were classified as high risk for surgery based on CPEX testing, 9 patients developed progressive/metastatic disease, 4 patients stopped treatment due to chemotherapy toxicity, 3 patients died whilst receiving treatment and 2 patients had incomplete records. Unfortunately, prior to 2015, our departmental records only recorded patients undergoing surgical resection for BR-PDAC and not those patients receiving NCR (Figure 1). Prior to 2015, 12 patients had undergone pancreatic resection for BR-PDAC, meaning that in total, 29 patients had undergone pancreatic resection for BR-PDAC following NCR. All these patients were treated with neoadjuvant chemotherapy followed by chemoradiation, as described in Section 2 above, prior to surgical resection. The regimens varied over the study period: FOLFIRINOX (*n* = 17), GEMCAP (*n* = 10) and gemcitabine and oxaliplatin (*n* = 2); see Table 1. Chemotherapy was followed by chemoradiotherapy in our patient cohort. Specifically, as described earlier, in our series, 12 resections were performed between 2010 and 2014 and 17 resections from 2015 to 2020 (Figure 1). All patients underwent radiological imaging during and at the end of treatment. For the purpose of analysis, the study groups were treated as the FOLFIRINOX group (*n* = 17) and the GEM group (GEMCAP and gemcitabine and oxaliplatin together (*n* = 12). As discussed later, our series had one peri-operative mortality that was included in the final analysis.

During NCR treatment, CA19–9 was measured at baseline at the time of diagnosis and prior to NCR treatment, at the end of NCR treatment and during follow-up after pancreatic resection. Figure 2 demonstrates the effects of NCR on serum CA19–9 levels. At baseline, patients had a mean serum CA19–9 level of 1696 IU/mL, and following NCR and prior to pancreatic resection, serum levels dropped to 69 IU/mL. The effects of recurrence on serum CA19–9 levels are discussed later.

### 3.3. Peri-Operative Outcomes for Patients Following Resection of BR-PDAC Following NCR

Of 29 patients, 6 patients underwent classical Whipple’s resection and 23 underwent PPPD. Concomitant venous resection was required in 18 patients (62%); of these, partial venous resection was required in 13 patients (44%) and full venous resection in 5 patients (17%). In patients requiring partial venous resection, repair was undertaken in all cases with interrupted non-absorbable sutures. In all patients requiring full venous resection, reconstruction was performed with primary end-to-end venous repair with splenic re-implantation, if required. No patients required arterial resection and/or other concomitant visceral resection. No frozen section revealed the presence of tumours at the pancreatic section margin (*n* = 19). There was a median blood loss of 519 mL (Table 3). Pancreatic anastomosis was performed using duct-to-mucosa anastomosis in 16 patients (55%) and using a dunking anastomosis in the remaining 13 patients (45%). In all cases, biliary reconstruction was performed with continuous absorbable sutures. There were no bile leaks noted in our series.

There was one post-operative death in our series. This 63-year-old female patient underwent PPPD after NCR (GEMCAP) in 2011. Intra-operatively, there was 2.2 L of blood loss and a full SMV resection was performed. The patient was haemodynamically stable at the end of surgery but required increasing inotropic support 6 h after surgery. Upon re-laparotomy, portal vein thrombosis was noted, and despite revision of the anastomosis, the patient died within 24 h of surgery. There was also one return to theatre for suspected bowel ischaemia after PPPD with full SMV resection and reconstruction. The small bowel was viable at laparotomy, with the patient being managed with laparostomy and then being discharged home 18 days following the primary surgery.

DFA was measured in 18 patients after resection (64%), with a median value of 42 IU/L (range < 30–96 IU/L). Aside from the one peri-operative death, no other 90-day mortality was noted. The Clavien-Dindo complication frequency is also presented in Table 3. Of note, 4 patients had wound complications that required negative pressure dressing to achieve resolution. Overall, the major morbidity for patients undergoing resection for BR-PDAC was 28% and mortality was 3%.

### 3.4. Histological Analysis

The median tumour size, as measured on the final histopathology after NCR, was 8 mm (4–13 mm IQR). R0 resection was achieved in 21 patients (73%) following surgery. In the FOLFIRINOX cohort, R0 resection was achieved in 94% of patients. No patients had positive pancreatic and/or bile duct resection margins. The SMV and posterior margin were the most frequently involved margins, with no patient having more than one positive margin. In 6 patients, a complete pathological response was noted following surgical resection (21%). Of note, perineural invasion was noted in 13 patients (45%) and vascular invasion in 10 patients (34%) and 7 patients had both perineural and vascular invasion (24%) present on post-operative histology. In addition, 4 patients received post-operative adjuvant chemotherapy (14%) (Table 4).

### 3.5. DFS and OS for Patients Post NCR

The median DFS for our whole cohort was 22 months but was significantly higher in the FOLFIRINOX group (34 months) compared to the GEM group (16 months); see Figure 3A. The median OS was 29 months for the cohort, with again the FOLFIRINOX group having a significantly higher median OS (42 months) compared to the GEM group (28 months); see Figure 3B. At the 5-year follow-up, 64% of the FOLFIRINOX group was alive compared to 0% of the GEM group. In total, 19 patients showed recurrent BR-PDAC at the end of the study period—5 patients showed recurrent BR-PDAC within the surgical resection site initially, with the remaining 14 having metastatic disease (9 liver metastatic disease, 3 peritoneal and liver disease, 1 liver and lung disease and 1 abdominal nodal recurrence). There was a clear link between serum CA19–9 levels and recurrent disease. The 10 patients who had not developed recurrence by the end of the study period had serum CA19–9 levels of 7 IU/L (range <2–13 IU/L), but in those patients who had recurrent disease, the serum CA19–9 level showed a clear trend of increase during the post-operative period, and at the time of a documented radiological recurrence, the serum CA19–9 level was 1818 IU/L (range 75–7049 IU/L).

## 4. Discussion

The aim of this single-centre retrospective study was to determine the DFS and OS following NCR for BR-PDAC. In patients with resectable PDAC, upfront surgery results in an R1 resection rate of 40–70% and 60–80% node positivity [4]. R0 resection is associated with superior patient survival following surgery for PDAC [5,12], and in patients with BR-PDAC, NCR can increase R0 resection rates with a concomitant increase in patient survival [19]. Furthermore, if a BR-PDAC tumour is considered resectable with venous resection, the risk of R1 resection remains [20]. In this context, NCR may enable vascular abutments to be sterilised, thus converting BR-PDAC to resectable tumours and increasing the rate of complete R0 resection, as shown in the AGEO study [21].

Early reports of resection for BR- and locally advanced (LA)-PDAC demonstrated a median PFS of 11.7 months and an R0 resection rate of 23% [22]. More recent prospective studies have shown improved survival, with Murphy et al. reporting that in patients who underwent resection, the median PFS was 48.6 months and the 2-year OS was 72% [23]. Our study demonstrated that NCR offers a good DFS and OS for patients undergoing resection for BR-PDAC, with our results also showing that the survival advantage is greater in those patients who receive FOLFIRINOX, which is consistent with previous studies [24]. In aggregate, the results of our retrospective study, in combination with other existing data, suggest a favourable rate of R0 resection in patients with BR-PDAC compared to prior non-FOLFIRINOX-based studies that have suggested R0 resection rates of approximately 40% [11,25,26,27]. There remain a number of ongoing FOLFIRINOX-based trials, such as NEPAFOX and NorPACT-1, that may demonstrate similar results in a controlled systematic setting. Overall, the R0 resection rate for our NCR cohort was 73%, but in patients who received NCR with FOLFIRINOX, the rate was higher at 94%. These results are comparable to recent studies and meta-analyses [28,29]. However, the reported R0 resection rates must be interpreted in light of differing definitions of R0: If R0 is interpreted as any margin > 0 mm, then rates of 77% are reported, and if R0 is interpreted as margins ≥ 2 mm, the reported rate declines to 29% [5,30].

The increased rates of R0 resection observed after NCR may be due to a reduction in tumour volume. [31]. Although previous systematic reviews have attributed no death to FOLFIRINOX, our study did demonstrate that patients either had to stop chemotherapy due to toxicity or had died whilst receiving treatment prior to surgical resection [32,33]. The additional discrepancies seen in the results reported after pancreatic resection after NCR can be partially attributed to the heterogeneous regimens used. For instance, in patients with BR-PDAC, Katz et al. provided patients with four cycles of FOLFIRINOX, followed by surgery and two cycles of adjuvant gemcitabine, whereas Murphy et al. provided patients with eight cycles of FOLFIRINOX, followed by radiation and adjuvant chemotherapy [11,23]. Katz et al. reported an R0 resection rate of 64% and a median OS of 21 months, whereas Murphy et al. demonstrated an improvement with their regimen [11,23]. In our reported series, BR-PDAC patients received more cycles of neoadjuvant chemotherapy than patients in these studies, with a further improvement in the R0 resection rate (73%) and in the median DFS and median OS. These data concur with the meta-analyses of Gillian et al. and Dhir et al. in demonstrating an improved OS for patients with BR-PDAC receiving neoadjuvant treatment [29,31]. Furthermore, Versteijne et al. conducted a meta-analysis of 38 studies of 3843 BR-PDAC patients and found a superior survival benefit of NCR compared to upfront resection, with a superior R0 resection rate (87 vs. 67%, *p* < 0.001). In addition, it is widely reported that the strongest prognostic factor after pancreatic resection is the lymph node status and positivity. Our study demonstrated that the post-operative rate of positive lymph nodes after NCR was 18% lower than that in other recent studies [24] and markedly lower than in patients who undergo immediate surgery (range 60–80%) [4]. Underscoring this is the recent Alliance data that further demonstrate an improved OS in patients receiving FOLFIRINOX-based neoadjuvant chemotherapy [34]. Interestingly, few patients in our cohort received adjuvant chemotherapy (14%) in contrast to the study reported by Katz et al. [11]. Given that 24% of our patients had both perineural and vascular invasion present on post-operative histology, this should be considered in patients after pancreatic resection, given the low levels of surgical morbidity in our series, and would be an improvement on the 55% patients able to receive adjuvant chemotherapy after upfront surgery [35].

Our post-operative mortality was 3%, similar to the published results of upfront pancreatectomy, and occurred in a patient at the beginning of our surgical experience with NCR resections [36,37]. Indeed, as demonstrated by our study, the use of NCR increased in our group, with 17 of the 29 pancreatic resections having been performed since 2015, and we envisage a greater role for this modality in BR-PDAC. As suggested by previous authors, we also believe that the reduced frequency of post-operative complications following pancreatic resection after NCR is related to the reduced level of pancreatic fistulas [38,39]. In those patients where DFA was able to be measured day 1 levels (*n* = 18), amylase levels did not exceed 100 IU/L.

The pattern of recurrence in patients was largely liver metastatic disease (47%). Noticeably, the local control rate was high in patients treated with NCR, with five local relapses noted without metastasis, whereas frontline surgery is usually associated with 35–86% of local relapse [39,40]. Furthermore, our study demonstrated an association between continued low levels of serum CA19–9 and the absence of radiological definable recurrent disease; conversely, in patients where recurrent disease was evident, there was a marked increase in serum CA19–9 levels that in some patients were elevated before the confirmation of radiological recurrence, as noted previously [41].

Finally, previous studies have reported that an NCR strategy achieved between 33 and 47% secondary resection rates [19,29], which is higher than that in our study, where the rate was noted to be 27%. In addition, there were limitations of this study, including a small patient population despite a study period of 10 years. The study also used different chemotherapy regimens, and there was no control group, such as an upfront surgery group. As such, whether NCR offers any advantage in terms of the DFS or OS to patients undergoing upfront surgery could not be assessed by this study, but the study does suggest that FOLFIRINOX appears to show improved survival benefits after surgery compared to GEM-based treatment when used as NCR for BR-PDAC.

## 5. Conclusions

In conclusion, our data confirm that an NCR regimen consisting particularly of FOLFIRINOX chemotherapy followed by chemoradiotherapy is an efficient strategy for patients with BR-PDAC, resulting in a good rate of R0 resection and improved DFS and OS.

## Figures and Tables

**Figure 1 cancers-14-04678-f001:**
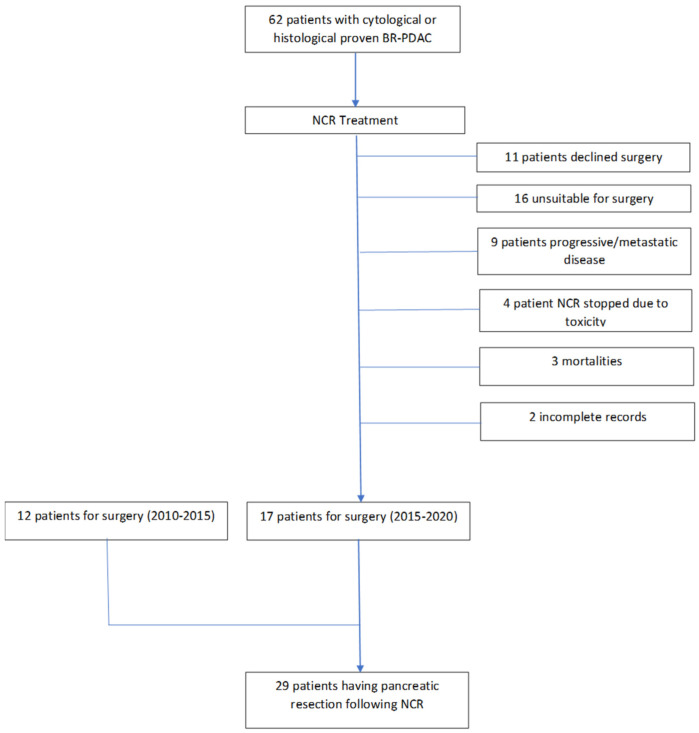
Flowchart depicting the patients’ cohort undergoing pancreatic resection for BR-PDAC.

**Figure 2 cancers-14-04678-f002:**
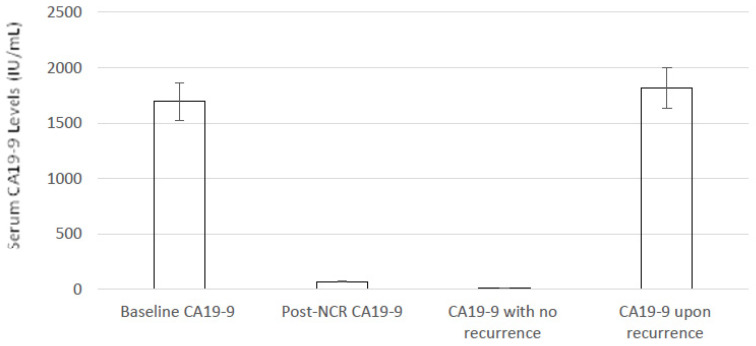
Serum CA19–9 levels in patients pre- and post-NCR for BR-PDAC.

**Figure 3 cancers-14-04678-f003:**
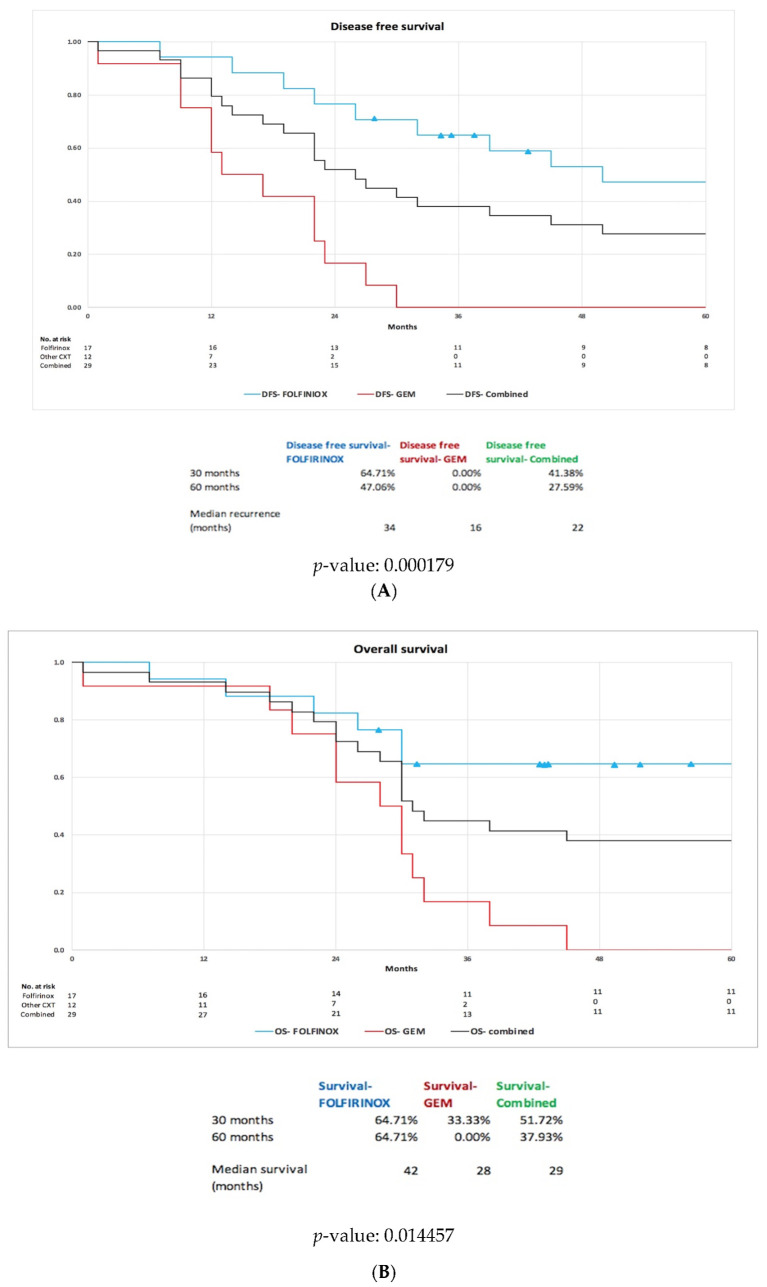
(**A**) DFS for patients after NCR and (**B**) OS for patients after NCR.

**Table 1 cancers-14-04678-t001:** Chemotherapy regimens for the study cohort.

Chemotherapy	Number of Cycles Median (Range)	*n*
FOLFIRINOX - Pancreatic head - Uncinate process	11 (8–12)	17 4 13
Capecitabine + gemcitabine - Pancreatic head - Uncinate process	11 (9–12)	10 7 3
Gemcitabine + oxaliplatin - Uncinate process	12	2 2

**Table 2 cancers-14-04678-t002:** Characteristics of patients receiving NCR.

Patient Characteristics	*n*	%
**Age**		
<65	12	41
>65	17	59
**Gender**		
Male	15	52
Female	14	48
**Performance status**		
0	18	62
1	11	38
**Diabetes mellitus**	7	24
**BMI**	25.5 (20–41)	-
**Pre-operative albumin**	37.4 (27–46)	-

**Table 3 cancers-14-04678-t003:** Intra-operative and post-operative complications occurring in patients undergoing pancreatic resection following NCR.

	Median (Range)	*n*	%
**Blood loss (mL)**	519 (300–2500)	-	-
**DFA**	42 (<30–96)	18	62
**Clavien-Dindo**			
**III**	-	6	21
**IV**	-	2	7
**V**	-	1	3

**Table 4 cancers-14-04678-t004:** Post-operative histological results.

Histological Results	*n*	%
**ypT0–1**	11	37
**ypT2-T3-T4**	18	62
**ypN0**	23	79
**ypN1**	6	21
**R0 resection** **FOLFIRINOX** **OTHER**	21 16 5	73 94 42
**R1 resection** **SMV margin** **Posterior margin**	8 4 3	27 14 10
**ypT0N0R0**	6	21
**Perineural invasion**	13	45
**Vascular invasion**	10	34
**Perineural + vascular invasion**	7	24

## Data Availability

Data are contained within the article.

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
