# Peer review of "Neoadjuvant Chemotherapy-Chemoradiation for Borderline-Resectable Pancreatic Adenocarcinoma: A UK Tertiary Surgical Oncology Centre Series"

_cancers, 2022, doi:10.3390/cancers14194678_

Round 1

Reviewer 1 Report (Previous Reviewer 2)

The manuscript has now been well-revised according to the reviewer's comments and queries.

Author Response

We thank the reviewer for their review and critique of our submitted manuscript. We thank the reviewer for their positive comments regarding our presented results.  

  • We thank the reviewer for their prior comments and are grateful that the reviewer’s comments and recommendations have been answered in full.

Reviewer 2 Report (Previous Reviewer 3)

In previous comment, I pointed out the following; the description of the survival curves in Figures 1 and 2 appears to be reversed for FOLFIRINOX and all cohorts.

However, it appears that this has not been corrected in the revised version.

Furthermore, the overall survival (OS) is generally longer than disease-free survival (DFS). This is because many pancreatic cancer patients die via cancer recurrence. However, in this cohort, DFS was longer than OS, especially except in the GEM group, which seemed quite strange. The authors must address this paradox adequately, or check the data carefully.

Author Response

We thank the reviewer for their comments regarding our submitted manuscript. In accordance with the reviewers comments we have made the following changes in the revised manuscript.  

  • We again apologise to the reviewer as we thought we had addressed the description of the survival curves in the previously revised manuscript. We have now performed the survival analysis for Figure 1 (DFS) and Figure 2 (OS) respectively again. These are now 2 new updated Figures with at risk data and the appropriate p values within the Figures. We have thoroughly checked in the revised manuscript and are confident it is now correct and will meet with expectations of the reviewer. We again apologise for this previous oversight.

Reviewer 3 Report (New Reviewer)

The paper is not original, describing the effects of the preoperative treatment in PDAC patients. Small sample size, long study period, different chemo, and the lack of control group are major concerns that should added a limits. However, the paper is well written. No more comments.

Author Response

We thank the reviewer for their comments regarding our submitted manuscript.

  • We thank the reviewer for their comments. We completely agree that the study has limitations as the reviewer details. In line with the reveiwer’s recommendations we have added the major limitations of our study in the revised discussion section. We hope that this addresses the reviewer’s comments adequately.

Reviewer 4 Report (New Reviewer)

Re: Neoadjuvant chemotherapy-chemoradiation for Borderline Resectable Pancreatic Adenocarcinoma: A UK Tertiary Surgical Oncology Centre Series General Comments: This report by Dr. Borbudhun is a retrospective review of outcomes of patients with borderline resectable pancreas ductal adenocarcinoma (BR-PDAC). In this analysis, patients were included from 2010 to 2020. All patients received neoadjuvant chemotherapy and radiation (NCR). Outcomes measured included overall survival (OS), morbidity, R0 resection rate, histological parameters, and disease-free survival (DFS). The hypothesis tested was there was a difference in outcome compared with no upfront therapy. Secondarily, the authors concluded that NCR (FOLFIRNOX over gemcitabine-based regimens) resulted in better outcomes. Major Comments: 1. This is a timely and frequently discussed subject in audiences ranging from small local tumor boards to large international symposia. Recent data from the ALLIANCE trial examining BR PDAC was recently reported. 2. Our group, similar to the authors, and the most of the world, likely agree that PDAC should be viewed as a systemic disease; and the benefits of a neoadjuvant approach to patients with BR PDAC need investigation. 3. While retrospective in nature, a notable strength of this study is that it is from a single institution - thereby likely optimizing consistency and standardization in surgical techniques, radiographic staging, pathology examination, and chemotherapeutics. But as the authors acknowledge, a multi-institutional approach with a larger sample size would increase the application of the prognostic tool. Specifically, the sample size of 62 total patients, over 10 years, with only 29 patients to examine, make conclusions difficult to interpret. 4. The authors mention using the 2019 staging algorithm – before 2019 what staging method was used? 5. Were intraoperative frozen section specimens ever examined? This too is a controversial topic in pancreas cancer surgery. Minor Comments 1. Which patients were selected for Gem/cis versus FOLFIRINOX? 2. Despite the low numbers, the survival curves are impressive. Could you you put the summary data for the survival curves and P values near the charts? 3. Page 11 has a typo, “were” instead of “where.” Same sentence has capitalization error.

Author Response

We thank the reviewer for their review and critique of our submitted manuscript.

  • In accordance with the insightful suggestions from the reviewer we have included the outcomes of ALLIANCE trial in the discussion section of our revised manuscript and related to the findings of these to our study.
  • We agree the author that PDAC should be viewed as a systemic disease and have reiterated this again in our manuscript.
  • We agree with the reviewer that there advantages to the study being single centre, but that multi-institutional data would make firmer conclusions possible.
  • With reference to the reviewers comments we have strongly emphasized that although there are 29 patients in study the number performed since 2017 make up the majority of the study group and that this is a treatment paradigm that we are using more in our institution such that we would expect the use of NCR to increase in the future at least in our institution.
  • We thank the reviewer for their comment regarding the 2019 NCCN staging algorithm. To clarify all cross-sectional imaging of included patients was reviewed by a dedicated HPB radiologist (GB) to ensure that all meet with the 2019 NCCN guidelines. This has been detailed in the Methods and Materials section of the revised manuscript and GB acknowledged in the Acknowledgments section.
  • We agree with the reviewer that frozen section is a controversial area and since 2017 the pancreatic resection margin is always sent for frozen histological analysis. This has been stated and detailed in the revised Methods and Results section.
  • As we have stated in the Methods section the decision for the type of neoadjuvant chemotherapy was based on MDT discussion but current practice centres on FOLFIRINOX as standard in our department. We have reiterated this in our revised Methods section.
  • In line with the reviewers request we have provided summary data and p-values in the revised DFS and OS Kaplan-Meier plots.
  • The spelling error and capitalization errors on page 11 have been correct and the manuscript have been checked for other spelling and grammatical errors

Round 2

Reviewer 2 Report (Previous Reviewer 3)

The authors addressed points raised by reviewers.

Author Response

We thank the reviewer for their review of our revised manuscript. As the reviewers suggested we have updated the statistical analysis with log-rank analysis. We have also updated patient numbers and percentages in the table and figures within the manuscript. The revised manuscript has also been checked for spelling and grammar. The changes have been highlighted in red text throughout the manuscript. We hope these changes meet the reviewers expectations and that our revised manuscript is of satisfactory quality for publication. 

This manuscript is a resubmission of an earlier submission. The following is a list of the peer review reports and author responses from that submission.

Round 1

Reviewer 1 Report

This is a single centre retrospective study, based on a case series, reporting short and long-term outcomes of patients undergoing PD following neoadjuvant chemo-radiotherapy, for borderline resectable pancreatic cancer.

29 patients undergoing FOLFIRINOX or gemcitabine (GEM) based NCR regimens, achieving R0 resection in 75% of cases. Median DFS and OS was 35 and 30 months respectively. 

The authors concluded that neoadjuvant chemo-radiotherapy for BR-PDAC resulted in higher rates of R0 resection and increased median DFS and OS, supporting its continued use in this patient group.

The authors need to be congratulated for their effort in investigating an important and still debated issue.

Some major concerns rose during revision and they should necessarily be addressed before considering the study for publication:

  • The authors reported that all patients had pre-operative biliary drainage. Considering the presence of patients affected by uncinate process cancer, in which jaundice is usually not so frequent, why was drainage always performed? And what kind of drainage: endoscopic or percutaneous?
  • Primary and secondary endpoints were not described in the methods.
  • Methods section is rather poor and statistics were not described.
  • The study presents a relevant selection bias: only patients which underwent surgery were considered; the entire cohort of patients receiving indication to CRT should be considered in order to take into account also cases of progression or death during CRT. 
  • No mention was reported about pathology protocol for specimen assessment. Please specify it in the methods.
  • The authors reported quite better results in patients undergoing FOLFIRINOX based regimen in comparison to those receiving gemcitabine based regimen; however the clinico-pathological characteristics of these two groups were not reported separately and were not compared to assess potential confounding factors

Reviewer 2 Report

The authors described a manuscript entitled “Neoadjuvant Chemoradiation for Borderline Resectable adenocarcinoma of the Pancreatic Head: A United Kingdom Tertiary Surgical Oncology Centre Study”, as a retrospective study. They concluded that neoadjuvant chemoradiation increased survival for borderline resectable (BR) pancreatic adenocarcinoma (PDAC). The manuscript has important information for readers in this field. I have several comments to improve it.

  1. This study included only for the patients undergoing pancreatoduodenectomy. However, the patients diagnosed BR-PDAC before treatment, might include some occult metastases, such as small peritoneal deposit. Please include the all patients with BR-PDAC planned resection (some of them might not undergo curative surgery) and add the patient’s flowchart. Resection rate should also be presented in the results because the neoadjuvant intervention might cause significant selection bias to confuse the implication of the treatment efficacy.
  2. Serial measurement of CA19-9 could provide useful information about the disease stage, aggressiveness, and response of the treatment. Please add the serial CA19-9 value (Pre-neoadjuvant and Post-surgery CA19-9) in addition to Pre-operative CA19-9, presented in Table 2.
  3. This cohort includes 6 cases (20%) diagnosed pathological complete response (pCR). The rate of pCR might be high compared with other study. Was histological proof of all patient confirm before surgery? How about above 6 cases?
  4. The color of lines in Figure 1 and 2 should be wrong. Please check them.

Reviewer 3 Report

This is a single-center, journalistic study of patients with BR-PDAC of the pancreatic head who underwent neoadjuvant treatment and subsequent surgery, showing preoperative treatment and postoperative survival outcomes.

I would first like to thank the editor for this peer-review opportunity.

Preoperative treatment for BR-PDAC is already common, and there are many reports of treatment outcomes. This study is an excellent cohort study in terms of high R0 resection rate, but the sample size is small and there are no new regulations that can be added to many existing studies. Therefore, its quality is inadequate to be treated in this journal.

Comment

In Figures 1 and 2, the explanation of the survival curve seems to be reversed for FOLFIRINOX and all cohorts.

Reviewer 4 Report

Small retrospective monocentric series on a 10 years period with very heterogenous treatment.

no clear message could be concluded from this study

Many series already published on this topic